# Did Italy Really Need Compulsory Vaccination against COVID-19 for Healthcare Workers? Results of a Survey in a Centre for Maternal and Child Health

**DOI:** 10.3390/vaccines10081293

**Published:** 2022-08-10

**Authors:** Michela Peruch, Paola Toscani, Nicoletta Grassi, Giulia Zamagni, Lorenzo Monasta, Davide Radaelli, Tommaso Livieri, Alessandro Manfredi, Stefano D’Errico

**Affiliations:** 1Department of Medical Surgical and Health Sciences, University of Trieste, 34149 Trieste, Italy; 2Institute for Maternal and Child Health-IRCCS Burlo Garofolo, 34137 Trieste, Italy

**Keywords:** compulsory COVID-19 vaccination, healthcare workers, vaccine attitudes, vaccine hesitancy, vaccine acceptance, Italian legislation, international legislation

## Abstract

Since its early spread, the COVID-19 pandemic has become a health threat globally. Due to their crucial role in the pandemic, Italy declared compulsory vaccination for healthcare workers. Vaccine hesitancy was observed among the healthcare workers and an ethical debate arose about Italian legal statement D.L. n. 44/2021. In this article, we present the results of a survey performed in an Italian center for maternal and infant care and assess the attitudes towards the COVID-19 pandemic and the mandatory COVID-19 vaccination of healthcare workers. Since March 2022, 91.5% of healthcare workers have been vaccinated with an additional dose. Only 2.3% of the respondents refused to take vaccination: the reasons behind this refusal were distrust, doubts over safety, and lack of information. Despite the high rate of response to vaccination, 17.7% of HCWs did not agree with its mandatory nature. In addition, 5.4% stated that they agreed to be vaccinated exclusively because of the sanctions provided for by the legislation. In conclusion, adequate vaccination coverage has been achieved in the hospital under consideration. However, it is still very important to continue to persuade HCWs of vaccine efficacy and safety, considering their social role.

## 1. Introduction

Since its early spread in late December 2019, COVID-19 has become a major public health threat globally [1]. According to the World Health Organization (WHO), which declared a state of pandemic in March 2020, as of 17 May 2022, more than 519 million cases and over 6.2 million deaths had occurred worldwide [2]. Overall, governments and healthcare systems applied a conservative approach, based mainly on non-pharmacological prevention measures: the use of facial masks became obligatory, the sanitization of locations and public transport was encouraged, and interpersonal relationships were reduced, thanks to measures ranging from simple social distancing to quarantine [3,4]. From December 2020 [5,6,7], in many countries all over the world, several vaccines were approved by EMA (European Medicines Agency) the Food and Drug Administration, for emergency use [8]. As of 15 May 2022, worldwide, more than 11 billion vaccine doses had been administered, and over 5.16 billion people had received at least one dose of a COVID-19 vaccine [2].

Due to their crucial role in the pandemic and their higher risk of contracting the infection, with significant morbidity and mortality, HCWs were the first group to be prioritized for vaccine distribution [9,10]. However, although the safety and effectiveness of COVID-19 vaccines have been clearly proven, skepticism and concerns about their reliability have become widespread, not only among the general population, but also among those who are expected to preserve health [3,11,12,13,14,15,16]. In the literature, specific COVID-19 vaccine hesitancy among HCWs ranges from 4.3% to 72% [17,18,19,20].

Italy was the first country in Europe to make vaccination against COVID-19 mandatory for HCWs [17,21,22,23]. On 1 April 2021, with the approval of Decree Law n. 44, the Italian government officially introduced the obligation for all health workers to be vaccinated against COVID-19 [24].

This paper aims to contribute to the debate on mandatory COVID-19 vaccination, and develops the topic on three levels. The primary purpose of our study is to identify the attitudes of HCWs related to COVID-19 vaccines, to study the phenomenon of vaccine hesitancy (without focusing on a specific type or brand of COVID-19 vaccine), and evaluate opinions on the mandatory vaccination of healthcare workers introduced with the D.L. n. 44, in the context of a center for maternal and child health. To achieve this, an analysis of the decree-law, compared with the European and international scenarios, will be presented for discussion.

## 2. Materials and Methods

To conduct this study, we developed a specific questionnaire, based on the items of greatest interest emerging from the most recent literature [25,26,27,28,29,30,31]. The questionnaire was then uploaded to the intranet network of a center for maternal and child health from 4 to 31 March 2022 and made available for voluntary compilation by the healthcare staff, which included an extensive list of professional figures. The number of subjects potentially responsive to the study was estimated to be 741 healthcare professionals, including 252 doctors, 328 nurses, and 161 other healthcare workers with various professional profiles. We sent a web link to access the questionnaire, including an introductory invitation explaining the objectives of the study, together with the assurance that the anonymity and confidentiality of the participants would be safeguarded.

The questionnaire was structured in four sections:
personal and professional characteristics (demographic data, professional profile, working environment with high or no infectious risk, state of health);perception of the pandemic (main sources of information, personal opinion of the pandemic’s impact on the population);anti-SARS-CoV-2 vaccination (perception of risks and benefits of vaccination, reasons for choosing or refusing to join the vaccination campaign);optional section (impact of the pandemic on the personal sphere).

The questionnaire was piloted on a sample of 20 subjects to determine its comprehensibility and the average completion time. Based on the feedback received, some items have been removed, while others have been merged. The research plan was preliminarily approved by the Hospital’s Institutional Review Board (IRB).

The collection and organization of data derived from the field questionnaires were carried out using Windows Excel software. After the preliminary phase of data homogenization, the calculation functions available in the software were applied. The results were presented graphically and with the aid of tables.

The results of the survey were described using frequency and percentages. Between-group differences were evaluated using a Chi-square test (or Fisher, when adequate). Graphical representations of the main results were performed using bar plots. Statistical significance was set at 0.05. The analyses were conducted using StataCorp, 2021, Stata Statistical Software: Release 17 (College Station, TX, USA: StataCorp LLC).

## 3. Results

We collected a total of 130 questionnaires filled out by the hospital’s health professionals during the study period. The overall participation rate was about 17.5%. Participation rates of 15% (38/252), 17.6% (58/32), and 21.1% (34/161) were observed in the category of physicians, nurses, and other HCWs, respectively. No questionnaire was excluded from the subsequent data-analysis phase.

The age range of the participants was homogeneous, with a clear prevalence of female subjects. Among the participants, 44.6% (*n* = 58) were nurses, 29.2% (*n* = 38) were physicians, while the remaining percentage included other professional profiles, in both the health and research fields. All the health workers stated that they did not have chronic diseases that could have contraindicated anti-SARS-CoV-2 vaccination. The results related to the personal data and professional profiles of the participants are summarized in Table 1, while Table 2 shows data related to their personal experiences with COVID-19. 

Regarding the perception of the pandemic, the sources of information were extremely heterogeneous (Figure 1). 

Overall, 39.2% (*n* = 51) of the participants stated that they had used more than one information channel, specifically scientific literature, social media, television programs, and scientific meetings. A total of 39.2% (*n* = 51) used only scientific and institutional sources, while 20% (*n* = 26) relied on the internet and social media, but always in combination with other sources. A total of 30% (*n* = 39) of the sample said the number of deaths attributable to COVID-19 was overestimated, but 89.2% (*n* = 116) concluded that complications from the infection could have a serious impact on people’s health. Without considering a specific factor, 60.8% of respondents (*n* = 79) believe that COVID-19 had a serious impact on the life of the entire population (Table 3).

When considering the merits of vaccinations, 53.1% (*n* = 69) of health workers said they were vaccinated annually against the flu virus and 42.3% (*n* = 55) always advised their patients to receive the recommended vaccinations, such as influenza vaccination in the case of people over 60. With specific regard to the anti-SARS-CoV-2 vaccine, 83.1% (*n* = 108) of the participants believed that the safety of a vaccine developed in emergency situations, and therefore rapidly, could be guaranteed. Overall, 96.1% (*n* = 125) of the respondents had been vaccinated and 91.5% (*n* = 119) of the total had completed the vaccination cycle with the third dose (Figure 2).

The great majority of the respondents (92.3%, *n* = 120) answered that the developed vaccines will be extremely useful to control the disease and to reduce any complications. In total, 90.8% (*n* = 118) of the participants said that they recommended vaccination to relatives and friends. However, 33.8% (*n* = 44) of the sample were concerned about short- and long-term vaccination complications (Table 4). 

Regarding the mandatory vaccination of health personnel, 82.3% (*n* = 107) of the participants believed that the introduction of this legislation was correct. Among those who chose not to be vaccinated and had no intention of doing so in the future (2.3%, *n* = 3), the reasons behind their decision were, in all cases, multiple, and concerned the safety of the vaccine (composition, side effects, possible reactions with pre-existing pathologies), the scarcity or conflict of the information received, the lack of trust in pharmaceutical companies and in the authorities responsible for their control, the belief that the disease is not serious, and that it could be easily controlled by physiological immunity (Figure 3). In one case, a mention was made of the scientific literature, which documented irreversible damage to the immune system following vaccination. Furthermore, among those who evaded the vaccination obligation, three participants declared that they had been suspended from the service without salary. Among the health workers who declared their opposition to vaccination, eight suggested alternative solutions for the containment of the pandemic: these measures essentially involved the correct use of personal protective devices, weekly SARS-CoV-2 tests for the entire population, the enhancement of territorial medical services, and the adoption of timely treatment in the event of infection. Most HCWs who were vaccinated or intended to be vaccinated in the future (96.2%, *n* = 125) cited multiple reasons: to access activities and services that would otherwise be precluded in the absence of Green Certification (bars, restaurants, cinemas, etc.); compliance with mandatory vaccination for healthcare workers; to reduce the likelihood of contagion or complications of the disease; and moral obligation towards patients. The motivation of only 5.4% of the subjects (*n* = 7) was exclusively linked to the legal obligation (Figure 4).

## 4. Discussion

The global health crisis resulting from the spread of COVID-19 has led to a debate on the ethics of compulsory vaccination for health workers [32,33,34,35]. The rationale for requiring HCWs to be vaccinated is dual: whilst it is the right of the health worker to be protected against occupational infections, on the other hand, there is a need to preserve the capacity of the health service and to protect patients themselves from being infected by operators [36]. However, the gap between the desired level of vaccination and reality has made it necessary to resort to making it mandatory [37]. 

On 1 April 2021, the Italian Government issued Decree Law n. 44, requiring that all HCWs, both in public and private institutions, be vaccinated against COVID-19 [38]. The professional figures involved were all HCWs who carried out their activities in social and healthcare institutions, public or private, and in pharmacies, parapharmacies, and professional studies. The control system consisted of several steps, each involving different institutions. Within no more than 5 days from the implementation of the decree, employers of health facilities were required to communicate the list of their members or employees, together with basic information, such as place of residence and region of reference. In the following 10 days, after carrying out some checks, the regions themselves reported to the local health authorities the names of the HCWs who were not yet vaccinated. On receiving this information, the local health authority asked the person concerned to produce documentary proof of vaccination within 5 days. Alternatively, they could provide documentary evidence of their right to exemption, which was only contemplated in the case of a “proven health risk”. In the absence of these requirements, no healthcare professional could be exempted because, as stated in the decree, vaccination was “an essential requirement for the practice of the profession”. The adoption of the measure of assessment led to the suspension of the right to perform services or tasks involving interpersonal contacts or, in any form, the risk of spreading the contagion from SARS-CoV-2. As a result, employers were required to assign duties to workers, whenever possible, that did not involve the risk of SARS-CoV-2 infection; failing this, immediate unpaid suspension was imposed until vaccination or, in any case, until the completion of the national vaccination plan. Regarding the health sector, Decree Law No. 44/2021 has since undergone further amendments and additions: currently, with Decree Law No. 24/2022 in force since 25 March 2022, the mandatory COVID-19 vaccination of health workers will remain active until 31 December 2022, entailing, in the event of non-compliance, the same types of measures as those described above [3].

In light of the Italian measures, several European countries decided to take similar actions [39], making the vaccine mandatory for HCWs and other categories of workers; these included Hungary [40], Greece [41], France [42,43], Poland [44], Latvia [45], and Germany [46]. In line with the Italian data, in all these countries, the professional figures involved were those in the health sector, although Greece, France, Latvia, and Hungary expanded the obligation to include civil protection workers, educators, and others. The sanctions provided for by these countries were all similar to those already stipulated in the Italian legal provision. In contrast with the European trend, in the UK [47] and the Czech Republic [48], mandatory vaccination was initially introduced and then repealed before coming into force.

Country-by-country references to the legislation and the respective specifications are shown in Table 5.

Beyond European borders, in the last third of 2021, other countries favored mandatory vaccination for healthcare professionals and non-healthcare professionals. In particular, some Australian states—such as Tasmania, New South Wales [49], Northern Territory, and Australian Capital Territory—have imposed vaccination for certain types of employment and community activities. Similarly, New Zealand, on 15 November 2021, with the COVID-19 public health response (vaccinations) order 2021 (LI 2021/94), made vaccination for COVID-19 mandatory for teachers, health professionals, prison staff, and port and airport workers [50].

Considering the points outlined above, HCWs’ hesitancy towards COVID-19 vaccination remains an important public health issue globally [51]. For example, studies carried out before the COVID-19 vaccines were distributed evidenced that vaccine acceptance in Italy was around 53.7% [52,53]. To our knowledge, most of the studies published investigate the acceptance of COVID-19 vaccines among HCWs by assessing intention rather than actual vaccine uptake [54,55,56,57]. Our study, on the contrary, was conducted in March 2022, when the state of emergency in Italy was coming to an end (decree law n. 24/2022) and almost a year had passed since the implementation of mandatory vaccination. For these reasons, we were able to ascertain the percentage of HCWs already vaccinated with a booster dose, which was found to be 91.5%. In January 2022, Shakell et al. [58] published a systematic review on COVID-19-vaccine acceptance: it emerged that Italian nurses had one of the highest acceptance rates (91.50%), which was in line with our data, considering that 87.9% of the nurses claimed to have been vaccinated with a third dose. Altogether, only three (2.3%) respondents, two nurses, and another health professional expressed their total refusal to take the vaccine, confirming, moreover, that they had been suspended from service without salary. In addition, two other nurses expressed their willingness to be vaccinated despite not having been vaccinated at the time.

These data are particularly interesting when compared with those relating to the flu vaccination [59]. In fact, only 53.1% of the health professionals reported that they were vaccinated against the flu annually. This is in accordance with what was stated in a large cross-sectional study conducted during May 2021, in Greece, in which COVID-19-vaccination acceptance rates exceeded influenza vaccination acceptance rates [54]. The motivation behind the discrepancy between these two trends—that is, the high response rate towards the anti-COVID 19 vaccine and the lower rate towards the flu vaccine—is to be found among the reasons that led the HCWs to accept the treatment. Particularly interesting is the fact that 5.4% of the respondents cited the mandatory aspect as their sole reason for being vaccinated; on the other hand, it is a cause for concern that only 26.2% believe that the moral obligation towards patients and the use of vaccines as a weapon to stop infections are sufficient reasons to vaccinate themselves [60,61].

In the literature, the main reasons for vaccine hesitancy are concerns about vaccine safety, efficacy, and potential side effects [18,62,63,64,65,66,67,68]. Regarding this, 33.8% of our sample, although vaccinated, raised concerns about possible short- and long-term complications. However, 83.1% of the respondents believed that the safety of a vaccine developed during an emergency can be guaranteed. From those who chose not to be vaccinated, there no unique answers were received, but the reasons always referred to doubts about the safety of vaccines, concerns about information, lack of confidence in the authorities, pharmaceutical companies, and the effectiveness of vaccines, and the belief that physiological is preferable to induced immunity.

As expected, the HCWs with an overall positive attitude to vaccination tended to promote vaccination among their patients. An encouraging fact that emerged from our study is that 90.8% said they had recommended COVID-19 vaccination to relatives and friends. A previous Italian national survey obtained even more encouraging data, with only 1.66% of the respondents not willing to recommend the vaccine to relatives [69]. Our results may have been linked to a biased selection.

In addition, 82.3% believed that mandatory vaccination for HCWs is a fair measure in the context of an unprecedented emergency, such as that of the COVID-19 pandemic. This contrasts with earlier studies, in which mandatory policies were deemed appropriate by less than half of the respondents [62].

Regarding the sources of information used, as has repeatedly emerged from the literature [5,69,70], they were heterogeneous and rarely referred to a single source: overall, 28.4% of the interviewees cited scientific literature and meetings, while another 26% claimed to glean information from television programs, the Internet, and social media. It was specifically the rise of online forums and social media platforms that facilitated the spread of misinformation: this could be connected with the fact that 30.0% of the HCWs, 32.8% of whom were nurses, thought the numbers of cases and deaths had been overestimated.

## 5. Conclusions

This study highlighted a vaccine hesitancy rate of 3.8% among our sample of HCWs from a hospital. The reasons behind this choice are in line with those previously described in the literature: distrust, doubts over safety, and lack of information were the main concerns. It is interesting that not all those who had been vaccinated were agreed with the mandatory aspect, to the extent that 17.7% (the majority of whom were nurses) believed that mandatory vaccination is not an adequate measure. As a result, 5.4% of the respondents had been vaccinated exclusively because of the sanctions provided for by the legislation.

Our results should be interpreted in light of the fact that only about 17.5% of the sample replied to the questionnaire (130 out of 741 HCWs).

Participation rates of 15% (38/252), 17.6% (58/32), and 21.1% (34/161) were observed in the categories of physicians, nurses, and other HCWs, respectively. We cannot provide a full explanation of the low response rate among all the categories of HCWs and we cannot assume that the non-respondents would have had a different opinion on mandatory vaccination.

Considering the total percentage of unvaccinated participants (2.3%) and of those who would not have been vaccinated if there had not been the obligation (5.4%), we can affirm that a more incisive information campaign in our context would have produced similar results [71,72].

In conclusion, adequate vaccination coverage has been achieved in the hospital under consideration. However, it is still very important to continue to persuade HCWs about vaccine efficacy and safety, considering their social role.

## Figures and Tables

**Figure 1 vaccines-10-01293-f001:**
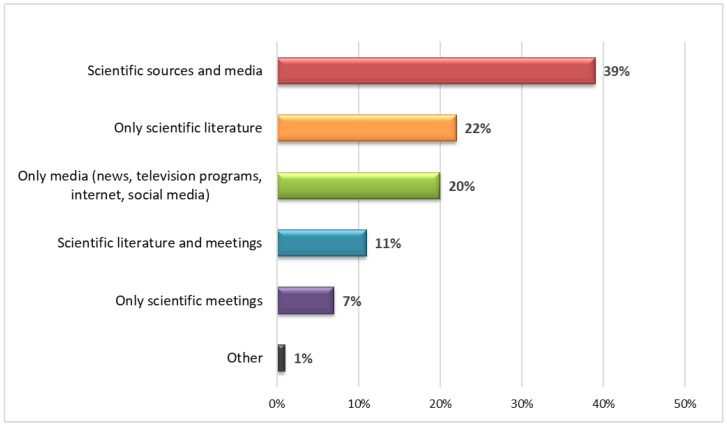
Main sources of information on the COVID-19 pandemic.

**Figure 2 vaccines-10-01293-f002:**
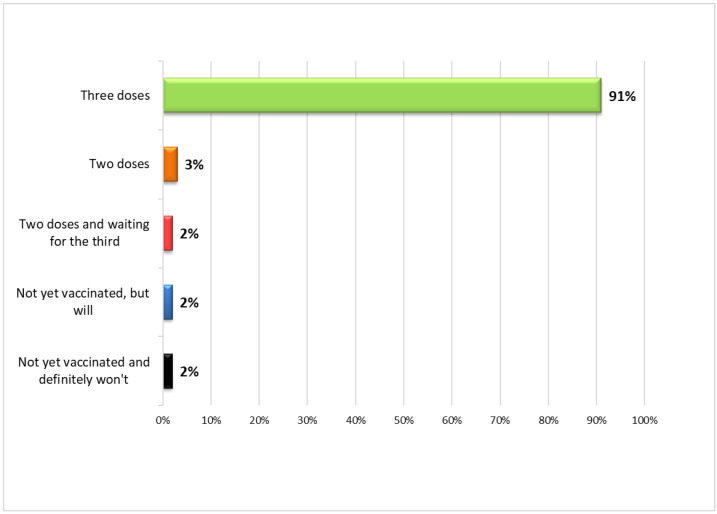
Vaccination status of healthcare workers.

**Figure 3 vaccines-10-01293-f003:**
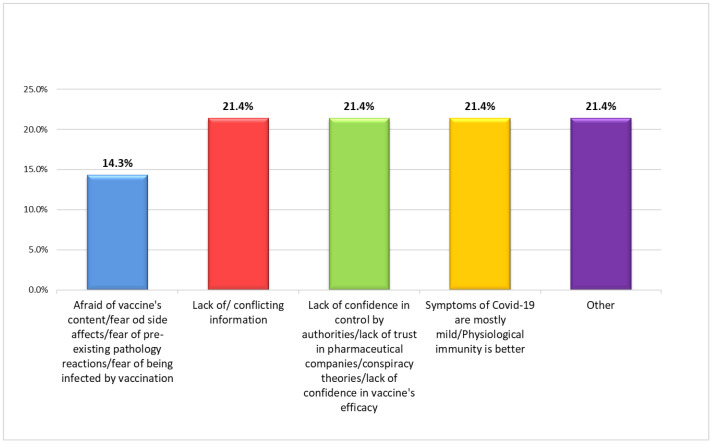
Reasons for vaccine refusal among respondents who declared their wish to not be vaccinated.

**Figure 4 vaccines-10-01293-f004:**
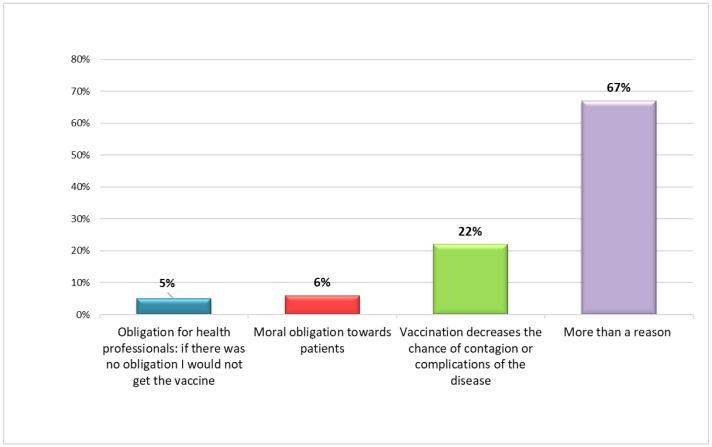
Main reasons for choosing to be vaccinated.

**Table 1 vaccines-10-01293-t001:** Demographic data, professional profiles, work-environment-related risk of infection, and health status of the participants.

Characteristic	Total Sample (*n* = 130)		Healthcare Workers/Profession						
			Physicians (*n* = 38)		Nurses (*n* = 58)		Other HCWs (*n* = 34)		*p*-Value
	*n*	%	*n*	%	*n*	%	*n*	%	
Age									0.192
</= 30	24	18.5%	4	10.5%	11	19.0%	9	26.5%	
31–40	29	22.3%	12	31.6%	12	20.7%	5	14.7%	
41–50	34	26.2%	12	31.6%	14	24.1%	8	23.5%	
51–60	39	30.0%	8	21.1%	21	36.2%	10	29.4%	
>60	4	3.1%	2	5.3%	0	0.0%	2	5.9%	
Sex									0.007
Male	33	25.4%	17	44.7%	10	17.2%	6	17.6%	
Female	97	74.6%	21	55.3%	48	82.8%	28	82.4%	
Does your profession put you in direct contact with patients?									<0.001
Yes	96	73.8%	34	89.5%	48	82.8%	14	41.2%	
No	34	26.2%	4	10.5%	10	17.2%	20	58.8%	
Have you worked in units with COVID-19 patients since the beginning of the COVID-19 emergency?									<0.001
Yes	75	57.7%	28	73.7%	37	63.8%	10	29.4%	
No	55	42.3%	10	26.3%	21	36.2%	24	70.6%	
Do you have a disease that prevented you from receiving the vaccine? (NB: This question takes into consideration one or more known conditions that have caused either your general practitioner OR the doctor present at the vaccination hub to deny you the possibility of being vaccinated)									na
Yes	0	0.0%	0	0.0%	0	0.0%	0	0.0%	
No	130	100.0%	38	100.0%	58	100.0%	34	100.0%	

**Table 2 vaccines-10-01293-t002:** Personal experiences of healthcare workers with COVID-19 infection.

Characteristic	Total Sample (*n* = 130)		Healthcare Workers/Profession						
			Physicians (*n* = 38)		Nurses (*n* = 58)		Other HCWs (*n* = 34)		*p*-Value
	*n*	%	*n*	%	*n*	%	*n*	%	
Have you tested positive for COVID-19 in the past?									0.634
*Yes*	46	35.4%	11	28.9%	22	37.9%	13	38.2%	
*No*	84	64.6%	27	71.1%	36	62.1%	21	61.8%	
Are you currently positive for COVID-19?									na
*Yes*	0	0.0%	0	0.0%	0	0.0%	0	0.0%	
*No*	130	100.0%	38	100.0%	58	100.0%	34	100.0%	
If you have tested/are currently positive for COVID-19, have you had/are you suffering from a form of infection that is:									0.438
Asymptomatic	6	4.6%	0	0.0%	5	8.6%	1	2.9%	
Mild symptomatic (common symptomatology of COVID-19 infection without the need for hospitalization)	31	23.8%	7	18.4%	13	22.4%	11	32.4%	
Severe symptoms (symptoms linked to COVID-19 infection with the need for assistance/hospitalization)	0	0.0%	0	0.0%	0	0.0%	0	0.0%	
Symptomatic with sequelae	1	0.8%	0	0.0%	1	1.7%	0	0.0%	
Are there people among your acquaintances (relatives and close friends) who tested/are currently positive for the COVID-19 test?									1.000
Yes	117	90.0%	35	92.1%	52	89.7%	30	88.2%	
No	12	9.2%	3	7.9%	6	10.3%	3	8.8%	
Are there people among your acquaintances (relatives and close friends) who died from COVID-19 infection?									0.498
Yes	30	23.1%	11	28.9%	11	19.0%	8	23.5%	
No	100	76.9%	27	71.1%	47	81.0%	26	76.5%	

**Table 3 vaccines-10-01293-t003:** Sources of information about the pandemic and its perception among the participants.

Characteristic	Total Sample (*n* = 130)		Healthcare Workers/Profession						
			Physicians (n = 38)		Nurses (n = 58)		Other HCWs (n = 34)		*p*-Value
	*n*	%	*n*	%	*n*	%	*n*	%	
What is your main source of information on the COVID-19 pandemic?									0.044
Only scientific literature	28	21.5%	11	28.9%	8	13.8%	9	26.5%	
Only scientific meetings	9	6.9%	2	5.3%	7	12.1%	0	0.0%	
Scientific literature and meetings	14	10.8%	6	15.8%	5	8.6%	3	8.8%	
Only media (news/television programs)/Internet and social media	26	20.0%	4	10.5%	14	24.1%	8	23.5%	
Scientific sources and media	51	39.2%	15	39.5%	23	39.7%	13	38.2%	
Other	2	1.5%	0	0.0%	1	1.7%	1	2.9%	
Do you think the number of cases and deaths has been overestimated?									0.050
Yes	39	30.0%	6	15.8%	19	32.8%	14	41.2%	
No	91	70.0%	32	84.2%	39	67.2%	20	58.8%	
Do you think that the complications derived from COVID-19 infection can have a serious impact on people’s health?									0.642
Yes	116	89.2%	35	92.1%	52	89.7%	29	85.3%	
No	14	10.8%	3	7.9%	6	10.3%	5	14.7%	
In your opinion, for the entire population, without delving into a specific area (health, economy, etc.), how serious is COVID-19 on a scale from 1 to 10?									0.430
Not severe (0–4)	7	5.4%	0	0.0%	5	8.6%	2	5.9%	
Moderately severe (5–6)	44	33.8%	13	34.2%	18	31.0%	13	38.2%	
Very severe (7–10)	79	60.8%	25	65.8%	35	60.3%	19	55.9%	

**Table 4 vaccines-10-01293-t004:** General attitudes of healthcare workers towards vaccinations (both recommended and anti-SARS-CoV-2), personal opinions about mandatory vaccination, and main reasons for joining or not joining the vaccination campaign.

Characteristic	Total Sample (*n* = 130)		Healthcare Workers/Profession						
			Physicians (*n* = 38)		Nurses (*n* = 58)		Other HCWs (*n* = 34)		*p*-Value
	*n*	%	*n*	%	*n*	%	*n*	%	
Do you receive the flu vaccination annually?									0.003
Yes	69	53.1%	29	76.3%	25	43.1%	15	44.1%	
No	61	46.9%	9	23.7%	33	56.9%	19	55.9%	
Do you advise your patients to receive the recommended vaccinations (e.g., anti-flu at > 60 years)?									<0.001
Always	55	42.3%	26	68.4%	23	39.7%	6	17.6%	
Sometimes	22	16.9%	3	7.9%	13	22.4%	6	17.6%	
Never	5	3.8%	0	0.0%	5	8.6%	0	0.0%	
It is not part of my professional duties	48	36.9%	9	23.7%	17	29.3%	22	64.7%	
Do you believe in science for the development of new, safe, and effective vaccines?									0.615
Yes	127	97.7%	38	100.0%	56	96.6%	33	97.1%	
No	3	2.3%	0	0.0%	2	3.4%	1	2.9%	
Do you believe that the safety of a vaccine developed during an emergency can be guaranteed?									0.008
Yes	108	83.1%	37	97.4%	44	75.9%	27	79.4%	
No	22	16.9%	1	2.6%	14	24.1%	7	20.6%	
Do you believe that the vaccine against the COVID-19 virus will be useful for the control of the disease?									0.027
Yes	120	92.3%	38	100%	50	86.2%	32	94.1%	
No	10	7.7%	0	/	8	13.8%	2	5.9%	
Are you concerned about the serious complications of the COVID-19 vaccine?									0.051
Yes, I’m seriously worried	9	6.9%	1	2.6%	7	12.1%	1	2.9%	
Yes, I’m worried	35	26.9%	6	15.8%	17	29.3%	12	35.3%	
No, I’m not worried	75	57.7%	24	63.2%	32	55.2%	19	55.9%	
No, I’m not worried at all	11	8.5%	7	18.4%	2	3.4%	2	5.9%	
Do you think that the mandatory vaccination of healthcare workers is right?									0.004
Yes	107	82.3%	37	97.4%	42	72.4%	28	82.4%	
No	23	17.7%	1	2.6%	16	27.6%	6	17.6%	
Did you received the COVID-19 vaccine?									0.623
Yes, I received two doses and the booster dose (third dose)	119	91.5%	38	100.0%	51	87.9%	30	88.2%	
Yes, I’m waiting for the third dose	2	1.5%	0	0.0%	1	1.7%	1	2.9%	
Yes, I received both doses	4	3.1%	0	0.0%	2	3.4%	2	5.9%	
No, but I definitely will	2	1.5%	0	0.0%	2	3.4%	0	0.0%	
No, and I definitely won’t	3	2.3%	0	0.0%	2	3.4%	1	2.9%	
If you have been vaccinated or are planning to be vaccinated, what are the reasons for your choice?									0.064
To have access to activities and services that would otherwise be precluded in the absence of Green Certification/Greenpass (bars, restaurants, cinemas, etc.)	0	0.0%		0.0%		0.0%		0.0%	
Obligatory vaccine for health professionals; if there was no obligation, I would not have vaccinated myself/I would not be vaccinated	6	4.9%	0	0.0%	5	9.6%	1	23.3%	
Vaccination decreases the chances of contagion or complications of the disease	27	21.9%	13	34.2%	9	17.3%	5	15.1%	
Moral obligation towards patients	7	5.7%	0	0.0%	4	7.7%	3	9.1%	
More than one option	83	67.5%	25	65.8%	34	65.4%	24	72.7%	
Do you recommend/have you recommended/will you advise your acquaintances (relatives and close friends) to be vaccinated against COVID-19?									0.023
Yes	118	90.8%	38	100.0%	49	84.5%	31	91.2%	
No	12	9.2%	0	0.0%	9	15.5%	3	8.8%	

**Table 5 vaccines-10-01293-t005:** European countries with mandatory COVID-19 vaccination for healthcare workers.

Country	Legal Reference	Come into Force	Professional Figures with Mandatory Vaccination	Population-Wide Mandatory Vaccination
France	Law 2021/1040, Articles 12–13	15 September 2021	HCWs, health professions students, fire, civil protection workers.	No
Germany	Infection Protection Act, Article 20a	15 March 2022	HCWs	No
Greece	Law 4829/2021, Article 206	12 July 2021	HCWs, firefighters.	Residents over 60 (Article 24, Law 4865/2021)
Hungary	Government Decree 449/2921 (VII.29.)	15 September 2021	Healthcare Education, cultural institutions, army (Government Decree 599/2921 (X.28.))	No
Italy	Decree Law N. 44/2021	1 April 2021	HCWs, police, education, social care	Residents over 50 (Decree-Law 1/2022)
Latvia	Amendments to the COVID-19 Infection Control Law	1 October 2021	Workers in private and public sectors (healthcare, education, etc.)	No
Poland	Dz. U. z 2022 r. poz. 340	1 March 2022	HCWs	No

## Data Availability

All data supporting reported results are availability from the corresponding author if requested.

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
