# Peer review of "Did Italy Really Need Compulsory Vaccination against COVID-19 for Healthcare Workers? Results of a Survey in a Centre for Maternal and Child Health"

_vaccines, 2022, doi:10.3390/vaccines10081293_

Round 1

Reviewer 1 Report

The objective of this study is to evaluate attitudes toward covid-19 vaccines and opinions about mandatory vaccination for healthcare workers. The authors give a very long and detail introduction about the history of SARS-CoV-2 infection in Italy and how the vaccination campaign was managed. The present the results of a short survey realized among healthcare workers in a Maternal and Child Hospital. Introduction and discussion were well written, but the data were analysed in a very simplistic way and did not support the conclusions.

.

Specific comments will follow

Introducion

1.      The introduction was clear, although too long and detailed and somehow redundant (“becoming the first country in Europe to do so” repeated twice, for example).

2.      Some concept, such as “variants” were mentioned without explanations.

Materials and methods

3.      Methods were presented with clarity although statistical methods were barely mentioned, even though tables reported p-values for test

4.      There should be consistency with the introduction in the use of COVID-9 and SARS-CoV-2 infection

5.      This survey could have benefit of already validated questionnaire about vaccine hesitancy

Results

6.      Results were poorly presented: after a short introduction, the authors presented only tables and figures, they did not guide the readers through the results.

7.      Please revise the titles of the tables: for example table 1, the title is “age and profession of the participants”, but more information besides age are given.

8.      Please avoid pie chart and use other type of plots, for example bar chart.

9.      Participation rate was poor, and it is acknowledged in the discussion as well as in the conclusion. But who are those who did not respond? May their answers be different from the responders?

Discussion

10.   The discussion was well written, but the main findings of the papers were reduced to three questions asked in the questionnaire and it was more an open discussion about the opportunity of the vaccines being mandatory among healthcare workers.

Conclusion

11.   The conclusions were a repetition of the discussion. As said before, data were stratify by working category: physicians were different from nurses and health care workers in many aspect and those who need to be persuaded about vaccines efficacy and safety are not all HCW, but only specific group of them.

Author Response

We are pleased to resubmit to the attention of the editorial board of Vaccines, the manuscript entitled “Did Italy Really Need Compulsory Vaccination Against COVID-19 for Healthcare Workers? Results of a Survey in a Centre for Maternal and Child Health” by Michela Peruch, Paola Toscani, Nicoletta Grassi, Davide Radaelli, Tommaso Livieri, Alessandro Manfredi and Stefano D’Errico. We thank the reviewers for their precious and kind suggestions. We performed all the changes proposed by the reviewers. The manuscript has been modified and all changes are in the text; below you can find the reply to each point raised by the reviewers.

Introduction

1 and 2.      The introduction was clear, although too long and detailed and somehow redundant (“becoming the first country in Europe to do so” repeated twice, for example). Some concept, such as “variants” were mentioned without explanations.

Thank you for your valuable suggestion. The introduction is now shorter and the text has been corrected by deleting repetitions. The concept of variants was deleted.

Materials and methods

3.      Methods were presented with clarity although statistical methods were barely mentioned, even though tables reported p-values for test
Thank you for your precious suggestion. The description of the statistical analysis carried out has been added.

  1. There should be consistency with the introduction in the use of COVID-9 and SARS-CoV-2 infection

This section has been reviewed according to the reviewer suggestions.

  1. This survey could have benefit of already validated questionnaire about vaccine hesitancy

Thank you for your valuable suggestion. Our questionnaire was conceived with all the authors according to other similar experiences in other countries

Results
6.      Results were poorly presented: after a short introduction, the authors presented only tables and figures, they did not guide the readers through the results.

Thank you for your valuable suggestion. Results are now better explained in the text so that the readers could be guided through the results.

7.      Please revise the titles of the tables: for example table 1, the title is “age and profession of the participants”, but more information besides age are given.

We really appreciated your suggestion. The titles of the tables have been corrected as suggested.

  1. Please avoid pie chart and use other type of plots, for example bar chart.

We are very grateful to the reviewer for this comment.

The graphics were reworked as suggested by the reviewer.

  1. Participation rate was poor, and it is acknowledged in the discussion as well as in the conclusion. But who are those who did not respond? May their answers be different from the responders?

Thank you for your precious suggestion. In the text the response rate was specified according to the different categories of HCWs (physicians, nurses, others). We don’t know why they didn’t answer and if their answers may be different from the reponders. 

Discussion

10.   The discussion was well written, but the main findings of the papers were reduced to three questions asked in the questionnaire and it was more an open discussion about the opportunity of the vaccines being mandatory among healthcare workers.

Thank you for precious comments. The aim of the authors was to investigate both aspects.

Conclusion

11.   The conclusions were a repetition of the discussion. As said before, data were stratify by working category: physicians were different from nurses and health care workers in many aspect and those who need to be persuaded about vaccines efficacy and safety are not all HCW, but only specific group of them.

Thank you for your valuable suggestion.  Discussion and conclusion were modified to respect your precious comments.

Reviewer 2 Report

General comments

This manuscript is an article about analysis of vaccination against COVID-19 for health care workers. This is a very well-described manuscript. However, there are some improvements that should be made. 

Specific comments

(1)Page14, Line310-311

Authors described that only about a quarter of the sample replied to the questionnaire (130 out of 500 HCWs).

What is the reason of no reply?

Please let me have author’s comment.

I hope that my comment is useful for the improvement of this manuscript.

Author Response

We are pleased to resubmit to the attention of the editorial board of Vaccines, the manuscript entitled “Did Italy Really Need Compulsory Vaccination Against COVID-19 for Healthcare Workers? Results of a Survey in a Centre for Maternal and Child Health” by Michela Peruch, Paola Toscani, Nicoletta Grassi, Davide Radaelli, Tommaso Livieri, Alessandro Manfredi and Stefano D’Errico. We thank the reviewers for their precious and kind suggestions. We performed all the changes proposed by the reviewers. The manuscript has been modified and all changes are in the text; below you can find the reply to each point raised by the reviewers.

(1)Page14, Line310-311. Authors described that only about a quarter of the sample replied to the questionnaire (130 out of 500 HCWs). What is the reason of no reply?

We don’t know the reason of no reply. But we added in the text more informations about the response rate which was stratified according to the different categories of HCWs (physicians, nurses and others) and this uncertainity has been explicitated in the text in the conclusions.

Round 2

Reviewer 1 Report

Lines 44-45: please revise

Author Response

Line 44-45 has been revised.